# Screening for Chagas Disease during Pregnancy in the United States—A Literature Review

**DOI:** 10.3390/tropicalmed6040202

**Published:** 2021-11-26

**Authors:** Elizabeth G. Livingston, Ryan Duggal, Sarah Dotters-Katz

**Affiliations:** Department of Obstetrics and Gynecology, Duke University, Durham, NC 27705, USA; ryan.duggal@duke.edu (R.D.); sarah.dotters-katz@duke.edu (S.D.-K.)

**Keywords:** Chagas disease, pregnancy, vertical transmission, maternal screening, congenital infection

## Abstract

Obstetrician-gynecologists in the United States have little clinical experience with the epidemiology, pathophysiology, diagnosis, and treatment of Chagas disease. The number of US parturients born in Central and South America has continued to increase over the last 20 years, making US obstetricians more and more likely to care for Chagas-infected mothers who may never be identified until dealing with long-term consequences of the disease. A literature search demonstrates that few US obstetric care providers recognize the risk of vertical transmission for the neonate and the missed opportunity of infant treatment to decrease disease prevalence. Most women will be asymptomatic during pregnancy, as will their neonates, making routine laboratory screening a necessity for the identification of at-risk neonates. While the benefits of treating asymptomatic women identified in pregnancy are not as clear as the benefits for the infants, future health screenings for evidence of the progression of Chagas disease may be beneficial to these families. The literature suggests that screening for Chagas in pregnancy in the US can be done in a cost-effective way. When viewed through an equity lens, this condition disproportionately affects families of lower socioeconomic means. Improved education of healthcare providers and appropriate resources for diagnosis and treatment can improve this disparity in health outcomes.

## 1. Introduction

Chagas disease is considered a “neglected tropical disease” by the World Health Organization (WHO), as it primarily affects poor populations living in tropical/subtropical conditions [1]. Due to immigration, obstetric care providers in the United States (US) are increasingly likely to care for pregnant women who are infected with Chagas. Although the US is not an endemic country, excepting some reported infections in the Southern States, vertical transmission is crucial to limiting the expansion of the disease among second-generation immigrants. Since 2007, with FDA approval of serologic assays for the diagnosis of Chagas, the United States Centers for Disease Control (CDC) has effectively focused on decreasing *T. cruzi* transmission via the screening of blood donations and donated organs. The CDC has also targeted mother-to-baby transmissions as a method to control the expansion of the disease [2].

A better understanding among obstetricians of Chagas infection, pathophysiology, epidemiology, screening techniques, and treatments is needed to suppress the spread of this disease. US Obstetrics residency education and professional society guidance on screening for and the care of women with Chagas disease are needed to reduce the occurrence of this disease in the US.

## 2. Transmission and Infection

Commonly, Chagas disease or American trypanosomiasis is transmitted to humans via an exposure to a triatomine insect infected with *Trypanosma cruzi*. The infected insects are generally found associated with poor housing conditions in Mexico and Central and South America, with only rare cases of insect transmission (autochthonous transmission) reported in the US [3,4]. The infection with *T. cruzi* occurs from a bite and contamination of a mucous membrane with the infected insect’s feces after its blood meal. These insects have been referred to as “kissing bugs” for the frequency of symptoms around the mouth and face. Not all infected persons will develop acute symptoms, and many of those who do will have generalized and nonspecific symptoms. During this acute phase, parasites can be identified in the bloodstream. The infection triggers a robust immune response but does not necessarily clear the infection [5]. If untreated, individuals may go on to the chronic phase of the disease, with 30% going on to develop cardiomyopathy and 10% developing digestive problems such as achalasia or toxic megacolon [6]. The lack of specific symptoms associated with acute infection, and the several-decade-long indeterminate phase of infection prior to symptoms, means that women of childbearing age are unlikely to know if they are infected with *T. cruzi* [7].

## 3. Epidemiology

Six to seven million people worldwide are thought to be infected with *T. cruzi* [6], with estimates of 300,000 individuals with infections living in the US. Forty thousand women of childbearing age living in the United States have chronic Chagas disease. The majority of US *T. cruzi* infections are due to the migration of infected individuals from Central and South America [8]. Estimates of the burden of the disease in the US suggest that 30,000–45,000 people in the US may have Chagas-related cardiomyopathy [9].

## 4. Congenital Transmission

Worldwide, the WHO estimates that 1,125,000 Latin American women of fertile age are infected with *T. cruzi*, with the delivery of 8600 cases of congenital chagas infection each year [10]. Congenital infection likely now accounts for 22.5% of new infections with *T. cruzi*, as insect-associated infections drop [10]. The estimates of US pregnant women infected with *T. cruzi* are around 6300 per year, with somewhere between 60 and 315 infections in neonates occurring via congenital transmission [8,11]. A study of cord blood donors in North Carolina showed an incidence of Chagas disease of 1.6/1000 with the serologic screening of cord blood donors in 2014 [12]. The risk of vertical transmission of Chagas in a chronically infected mother is estimated at 1–5% by the CDC [13] and was estimated at 4.7% in a 2014 metanalysis [14]. The transmission rate is likely higher in endemic countries than nonendemic countries [14].

Aside from coming from a country or region with a high prevalence of Chagas, other proposed risk factors include: a prior infected child or the mother has detectable parasitemia or a decreased T-cell response to *T. cruzi* and a coinfection with malaria or HIV [15]. The mechanism of vertical transmission is thought to be transplacental and is considered to occur more commonly in the second and third trimesters. Placental histopathology has been shown to identify the parasite [16].

Transmission is likely via breaks in the intervillous space or via marginal sinuses, though fetal oral ingestion is also speculated [15]. As in other infections with vertical transmission, acute infections with *T. cruzi* with high maternal parasite loads have been associated with higher rates of congenital infections [15]. Ten to forty percent of infected newborns will have symptoms such as low birth weights and hepatosplenomegaly and, less commonly, cardiac, meningoencephalitis, or hematologic problems [17]. These findings are nonspecific, and Chagas may not be suspected. Breastfeeding by the infected mother is generally considered safe, with admonitions to avoid nursing if nipples are cracked or bleeding [2].

## 5. Effect on Pregnancy

Generally, asymptomatic chronic Chagas disease is not thought to affect fertility, and if no vertical transmission occurs, pregnancy outcomes are thought to be similar to background risks [15]. Acute infection may be associated with spontaneous abortion. For infected fetuses, there are increased rates of preterm births, low birth weights, preterm premature rupture of the membranes, and hydrops [18,19].

The normal immunosuppression of pregnancy has been speculated to increase parasitemia in chronically infected women with Chagas [15,20]. While little is known on the impact of intercurrent pregnancy on the course of Chagas disease in infected women, pregnancy is generally not thought to hasten or worsen the disease outcomes [21].

## 6. Testing and Screening

Since most maternal infections will be asymptomatic and only 10–40% of newborns will demonstrate nonspecific symptoms, a more systematic approach to the identification of Chagas disease is needed for mothers and babies. The diagnosis of acute Chagas disease, as with congenital Chagas disease, is made through the identification of trypomastigotes in the peripheral blood through PCR testing, preferably, or Giemsa stain (less sensitive than PCR) [17]. As the techniques improve to overcome the limitations in sensitivity and specificity, PCR is becoming an important tool for the diagnosis of Chagas. Challenges in developing a reliable PCR include genetic variability with several types and subtypes of *T. cruzi* and cross-reactions with related infections such as Leishmania species [22]. The sensitivity of PCR techniques (especially in patients in the asymptomatic phase of Chagas) varies considerably, as negative PCR results do not exclude the presence of an infection, and a positive PCR assay is not pathognomonic of Chagas disease [22]. Consequently, a chronic disease diagnosis relies on serologic screening, usually two separate testing methods (involving distinct antigen sets) done in a cascade fashion. Individuals from different geographical areas may be infected with differing serotypes of *T. cruzi*, leading to differing sensitivities and specificities for various diagnostic tests. Some of the later generation diagnostic tests are not available commercially in the US. The CDC offers free validated confirmatory testing, which requires 10–14 days [23,24].

The screening strategies during pregnancy vary. The WHO and Pan American Health Organization recommend screening pregnant women in Latin America by incorporating this into other screenings during prenatal care [6,25]. The CDC recommends screening at-risk pregnant women who have lived in endemic areas [13]. A Boston-based clinic has reported using a two-question screening during the initial prenatal visit: (1) Where were you born? (2) Have you ever spent more than 6 months in Mexico/Central or South America? This two-question-based initial screening followed by serologic screening diagnosed 3/619 (0.5%) as Chagas-positive (the estimated incidence in the population) and allowed linkage to infectious disease care for mothers and infants [26]. Edwards and Montgomery, in a 2021 review, described the components of a comprehensive pregnancy-based screening program for congenital Chagas disease, utilizing features drawn from a successful program in Catalonia, Spain [24]. They recommended screening by history and testing at-risk mothers with a two-step serological-based method (see Appendix A). A referral for post-delivery/breastfeeding treatment should be available. Congenital infections can be diagnosed by the identification of *T. cruzi* in the infant’s blood (<three months) by PCR or Giemsa stain or serologies in an older infant [2]. Younger infants with positive PCR testing for *T. cruzi* should have the testing repeated for confirmation. Positives should be treated. Infected infants can be treated with benznidazole or nifurtimox, with a cure rate of 90% in the first year of life. Serologic testing should be performed of the negatives at 9–12 months to confirm the sero-reversion. Family members, including other children, maternal siblings, and parents, should be tested and referred for care and treatment, including EKGs and medication [24]. An alternative to maternal screening is cord blood screening for the presence of antibodies, which would reflect the maternal disease status from placentally transmitted IgG. The delay in return of a newborn test for congenital Chagas may interfere with the diagnosis and treatment if the mother and infant are lost to follow-up after delivery [7]. Screening with placental pathology has a low sensitivity [27].

## 7. Cost-Effectiveness

While neonatal screening for inherited diseases is routine in all 50 states [28], Edwards and Montgomery pointed out that, of the 33 tests recommended by the US Recommended Uniform Screening Panel (RUSP), congenital Chagas is more common than half of the disorders screened for in those panels [7]. Several investigators have indicated that the screening and treatment for Chagas disease will be cost-effective, with a savings of $1314/birth and $670 million of lifetime savings per birth year cohort [29]. Stillwagon et al. performed a decision analysis and found that universal maternal screening, infant testing, and treatment were cost-saving at a maternal prevalence greater than 0.065 and congenital transmission over 0.001% compared to no screening and estimated $638 million saved for every birth year cohort. The savings persisted even with the evaluation of false positives in low-prevalence regions [11].

Stillwagon et al. pointed out the hazards of targeted screening being a burden placed on obstetric care providers who identify at-risk patients based on last names or the location of birth [11]. Targeted screening may create a barrier to care if pregnant women from Latin America perceive that they are being singled out for a particular disease state.

## 8. Care after Diagnosis

For chronically infected asymptomatic mothers, the recommendation is that treatment be delayed until after pregnancy and breastfeeding [30]. There are case reports of Chagas treatment in the setting of undiagnosed pregnancy [31,32], without harm to mother or infant, though without consistent protection from vertical transmission. Treatment for Chagas disease, outside of pregnancy, is recommended for all infected individuals up to age 18 and is encouraged up to age 50 for those without cardiac involvement [2,30]. Treatment is not as effective in chronically infected individuals as for acute infections. For postpartum women, antiparasitic therapy with benznidazole or nifurtimox is administered for a prolonged course, at least 60 days. Side effects of the treatment are common in adults—gastrointestinal complaints (including anorexia, weight loss, and nausea and vomiting) occur in 30–70% of adults. Neurological complaints, both central and peripheral, occur: CNS irritability, insomnia, and, rarely, tremors or possible dose-dependent peripheral neuropathy seen late in the treatment course (up to 30%). Skin reactions (photosensitivity and, rarely, exfoliative) occur in 30% of adults. Bone marrow suppression or drug hepatitis are less common [30]. Drug-adverse events can be managed by temporarily stopping and restarting or halving and then up-titrating medications [33]. Twenty to twenty-five percent of those on medications will stop them due to side effects [34,35]. For chronically infected adults, there are a lack of useful assays to measure the clinical responses to therapy. Ten to forty percent of chronically infected individuals will eventually (at times, requiring up to 10–20 years) experience sero-reversion with treatment [30]. While short-term clinical markers of therapeutic responses are lacking, research-based PCR techniques suggest that 80% of subjects may clear *T. cruzi* with optimized regimens [36]. Clinical outcomes tend to be better in individuals treated earlier in the course of the disease [37]. Infectious disease consultation is recommended for the initiation of therapy, and assistance through the CDC is available [2]. Baseline labs with CBC and chemistries are recommended [30], with a q. 2–4-week follow-up to monitor side effects associated with therapy. For individuals diagnosed with Chagas, EKGs at regular intervals (every 1 to 2 years) are recommended, and individuals are warned to report GI symptoms. For infants diagnosed with congenital Chagas, treatment is indicated as soon as infection is confirmed, as treatment is particularly effective in the first year of life, with a cure rate as high as 90% [27]. Infants and children have lower rates of side effects from therapy [27]. Benznidazole is FDA approved for children as young as two years old, but a treating physician may opt to use it on younger children. As of 2020, Nifurtimox has FDA approval to use at birth for weights over 2.5 kg [7] (see Appendix A). While recommendations suggest treatment be delayed until after lactation, there is minimal nifurtimox or benznidazole exposure in breast-fed infants of treated mothers [38,39].

Strategies to prevent vertical transmission are limited. In a 2017 historical cohort study of 67 women in Argentina, 33 treated with benznidazole prior to conception had 0/42 infants with congenital Chagas, while the 52 untreated women had 16/114 infected children (14%) [40]. Among the cohort of previously treated mothers, 32% experienced sero-reversion over 7 years [40]. There are many new compounds, repurposed pharmaceuticals (used alone or in combination) and naturally occurring compounds, that have demonstrated anti-trypanosomal activity in vitro, in animal models and, occasionally, in humans [41,42]. Some might have potential for use later in pregnancy, such as novel azole compounds, while others may be contraindicated in pregnancy [41,42]. A suggested novel strategy for treating pregnant women is the use of therapeutic vaccines for Chagas disease [43]. While vaccines appear promising in murine models, therapeutic vaccines are not yet commercially available. A computational decision model suggests that, even with modest efficacy, a moderately priced vaccine would be cost-effective by preventing vertical transmission [43].

Recommendations on modifications to obstetric care after the diagnosis of Chagas disease in pregnancy are limited. Suggested care from a US center embarking on a Chagas pregnancy screening protocol recommends a baseline EKG and echocardiogram with repeat EKG in the late third trimester [44]. They additionally proposedgrowth scans in the third trimester and antenatal monitoring with nonstress tests and fluid checks starting at 32 weeks [44].

## 9. Health Care Equity and Chagas Disease

Only about 11% of US blood donors diagnosed with Chagas disease pursue treatment. This is generally thought to be due to immigrants from Latin America being marginalized, poor, and uninsured [45]. A study using focus groups in Georgia on Chagas disease among people from Latin America demonstrated little awareness about Chagas disease, economic limitations in seeking care, fear of mainstream medicine, and the use of alternative providers [45], though it was noted that mothers quickly sought out mainstream care when their children were ill [45]. Given the disproportionate burden that Spanish-speaking individuals carry with Chagas disease, screening and treatment are equity issues and quality issues in American medicine [46].

## 10. Educational Gaps for Obstetricians

There is little information available to practicing obstetricians on pregnancy and Chagas disease from the usual professional sources. Chagas disease is only mentioned briefly in a single American College of Obstetrics and Gynecology (ACOG) Practice Bulletin and only in the context of women being transfused for postpartum hemorrhage [47]. Chagas disease is indexed once in a leading Obstetric textbook, “Williams Obstetrics 25th ed.”, and then only as a potential cause of fetal hydrops [48]. Chagas is only indexed as a cause of achalasia in another leading textbook [49]. Recently, The Infectious Disease Society of Obstetrics and Gynecology has presented abstracts related to Chagas disease in 2019 and 2021 [26,44], suggesting interest inthe infection is increasing in this community. Aside from these abstracts, Chagas is rarely a topic of published articles in the most commonly read US obstetrics journals, with a 2005 citation in the *American Journal of Obstetrics and Gynecology* and a 1995 citation in *Obstetrics and Gynecology* [50,51]. In 2010, a survey of 400 US obstetricians on the knowledge of Chagas disease showed that 60% considered their knowledge on Chagas as limited; only 8% knew it could be transmitted vertically, and 70% never considered the diagnosis in women from endemic countries [52]. This lack of knowledge is unfortunate, as the number of US parturients who were born in Central and South America has continued to increase over the last 20 years, making US obstetricians more and more likely to care for Chagas-infected mothers, with over 6000 infected women delivering annually [8]. While US obstetricians have little experience with screening for Chagas disease, this is in contrast to the extensive guidelines for the screening and care of HIV-infected pregnant women [53,54]. The number of US cases of vertical Chagas transmission is similar to the estimated 5000 HIV-infected women delivering 73 HIV-infected neonates in 2016 [55]. Additionally, obstetricians quickly became accustomed to screening for Zika exposure as concern grew in 2016 to 2017 about the risk to fetuses. In 2018, the vertical transmission of Chagas was likely greater than the total number of Zika infections in the US (74) and territories (148) [56].

Contributing to US obstetricians’ lack of recognition of Chagas is the fact that the majority of Chagas-infected women will be asymptomatic during pregnancy. These infected women will likely remain asymptomatic for decades, but eventually, one-third of them will develop cardiac or gastrointestinal complications [57]. Identification of the infection will allow therapeutic options, *T. cruzi* baseline assessments with EKGs, and patient education on the manifestations of the disease if the Chagas disease progresses.

While obstetricians rarely recognize risk factors, signs, or symptoms of Chagas disease in mothers, the neonatal symptoms are also nonspecific: fever and anemia neonatal depression. This makes Chagas easily confused with other congenital infections, such as cytomegalovirus. Certain neonatal manifestations are less dramatic than those found with the vertical transmission of Zika and HIV, which may account for the lack of obstetric screening for and awareness of Chagas in the US.

The Centers for Disease Control recommends that the prevention of mother-to-baby transmission is an important control strategy for reducing US Chagas disease infections (in addition to screening blood and organ donations) [13]. Clearly, the treatment of identified newborns exposed to *T. cruzi* by their mothers in utero is highly efficacious, with treatment before age one being 90–100% effective [30]. While the treatment of chronically infected adults is less successful, the identification of infected mothers may provide some future health benefits for them personally, as well as for their current neonates and future children. Universal screening avoids obstetricians needing to profile patients based on their country of origin or surname [11].

## 11. Conclusions

In summary, to decrease the incidence and morbidity of Chagas disease in the US, the vertical transmission of *T. cruzi* should be addressed. For this to occur, obstetricians and other maternal health providers need an increased awareness of the disease through professional education and visible publications in high-profile obstetric journals. The obstetric professional societies should draft clear guidelines on the appropriate screening, treatment, and follow-up of pregnant women with Chagas disease and their offspring. The universal screening of pregnant women with Chagas disease may be an increasingly cost-effective approach as improved screening tests and treatments become available.

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
