# Peer review of "Screening for Chagas Disease during Pregnancy in the United States—A Literature Review"

_tropicalmed, 2021, doi:10.3390/tropicalmed6040202_

Round 1

Reviewer 1 Report

The review does not bring to the reader any news on thefield. It should be interesting if a kind of algorithm with the optimal treatment  time and dose of anti-trypanosoma drugs for the newborn. Also, a suggestion for mother treatment's. Concerning the PCR assay it should be emphasized that it is becoming an important tool for the diagnosis of Chagas disease. However, the sensitivity of PCR techniques (especially in patients in the asymptomatic phase of CD) varies considerably as negative PCR results do not exclude the presence of infection, and a positive PCR assay is not pathognomonic of CD.

Author Response

Response to Reviewer 1

Thank you for your review and comments.

“The review does not bring to the reader any news on the field.” This was intended to be a literature review with a focus on obstetric care and Chagas disease. There is little in the US Chagas literature written with an obstetric readership in mind.

“It should be interesting if a kind of algorithm with the optimal treatment  time and dose of anti-trypanosoma drugs for the newborn. Also, a suggestion for mother treatment's.”  I have created an algorithm in Figure 1 on the screening and care of pregnant women and their infants. I have enlarged the treatment section (8) to incorporate more information on treatment and monitoring.

“Concerning the PCR assay it should be emphasized that it is becoming an important tool for the diagnosis of Chagas disease. However, the sensitivity of PCR techniques (especially in patients in the asymptomatic phase of CD) varies considerably as negative PCR results do not exclude the presence of infection, and a positive PCR assay is not pathognomonic of CD.”  I have added more information on the benefits and drawbacks of PCR testing for Chagas.

Reviewer 2 Report

Thank you for the opportunity to revise the manuscript sent by Dr. Livingston et al about the screening for Chagas disease during pregnancy. As the authors mentioned, Chagas disease is a neglected disease affecting more than 6 million people worldwide and due to migration flows, the burden of the disease is increasing in non-endemic countries. In this setting, stopping vertical transmission is crucial and represents an opportunity to identify infected people and avoid new infections. Thus, protocols to handle pregnant patients are a need, and knowledge of the disease should be ensured for such practitioners. In this line, I really appreciate a review to highlight the importance of this problem and the opportunity that represents. To better understand the manuscript and recommend its acceptance for publication, I would like to ask for some clarifications:

  1. Introduction:

Line 28: This statement is not supported by any reference. Please, consider beginning with the next sentence.

Line 36: I would suggest emphasizing the fact that (excepting some reported infections in the southern states), the US is not an endemic country so vertical transmission is crucial limiting the expansion of the disease among 2nd generation immigrants.

Line 39: I would consider changing “eradicate” to “control its expansion”

  1. Congenital transmission:

Line 93-94: This information is further explained in section 6. I recommend not stating the same information repeatedly.

  1. Care after diagnosis:

Line 157-158: please, be more specific on side effects appearance: its incidence and sort them by frequency or importance.

Line 158-159: what do you mean by clearing the disease? One of the major problems is how we measure cure in Chagas disease. Although antibody negativization is the only accepted criteria, probably cure rates are higher. Please, state the difficulties of measuring this cure and mention that cure achievement is a progressive process that can last years or decades.

Line 167-169: Your statement of how treatment could avoid vertical transmission is too vague. There is evidence that treatment with benznidazole in childbearing age mothers drastically reduces congenital infection.

Álvarez MG, Vigliano C, Lococo B, Bertocchi G, Viotti R. Prevention of congenital Chagas disease by Benznidazole treatment in reproductive-age women. An observational study. Acta Trop 2017; Oct;174:149-152

Line 170:173: therapeutic vaccines are in pre-clinical stages and other drugs are also under development. Why did you consider mentioning only this measure?

References:

I would recommend avoiding CDC references so many times. Please, try to use the original cites used by CDC to generate their recommendations. Furthermore, please use references outside the US where many programs and experience in this field have been already generated.

Author Response

Response to Reviewer 2

Thank you for your thoughtful and thorough review- We believe your suggestions have enhanced the paper.

  1. Introduction:

Line 28: This statement is not supported by any reference. Please, consider beginning with the next sentence. The first sentence was deleted as suggested.

Line 36: I would suggest emphasizing the fact that (excepting some reported infections in the southern states), the US is not an endemic country so vertical transmission is crucial limiting the expansion of the disease among 2nd generation immigrants. We have added emphasis to the importance of limiting vertical transmission

Line 39: I would consider changing “eradicate” to “control its expansion”  The wording  has been changed. .

  1. Congenital transmission:

Line 93-94: This information is further explained in section 6. I recommend not stating the same information repeatedly. We have moved the portions on transmission to section 6 on testing and screening as recommended.

  1. Care after diagnosis:

Line 157-158: please, be more specific on side effects appearance: its incidence and sort them by frequency or importance. This section has been expanded to include more details on adverse events associated with treatment.

Line 158-159: what do you mean by clearing the disease? One of the major problems is how we measure cure in Chagas disease. Although antibody negativization is the only accepted criteria, probably cure rates are higher. Please, state the difficulties of measuring this cure and mention that cure achievement is a progressive process that can last years or decades. We have modified this section to provide more nuance on monitoring the effects of treatment.

Line 167-169: Your statement of how treatment could avoid vertical transmission is too vague. There is evidence that treatment with benznidazole in childbearing age mothers drastically reduces congenital infection.

Álvarez MG, Vigliano C, Lococo B, Bertocchi G, Viotti R. Prevention of congenital Chagas disease by Benznidazole treatment in reproductive-age women. An observational study. Acta Trop 2017; Oct;174:149-152 We added more specific details about the Alvarez  study in the text and highlighted the findings.

Line 170:173: therapeutic vaccines are in pre-clinical stages and other drugs are also under development. Why did you consider mentioning only this measure? Given currently available medications are not recommended in pregnancy, the use of a therapeutic vaccine seems innovative. We found the Bartsch paper utilizing a computational decisional model for prevention of vertical transmission with a vaccine hopeful for an effective, safe intervention. With the expanded discussion of the Alvarez paper utilizing drugs., there is more balance to the section. I have added a comment and reference about drugs in development for Chagas. I could identify no particular medication that has been investigated as a therapeutic for pregnant women or for targeted prevention of vertical transmission.

References:

I would recommend avoiding CDC references so many times. Please, try to use the original cites used by CDC to generate their recommendations. Furthermore, please use references outside the US where many programs and experience in this field have been already generated.

We have added original references in place of the CDC references. Though for clinical guideline recommendations, we have left the CDC references.